



# A Conceptual Framework for Integration Development of GSFLOW Model: Concerns and Issues Identified and Addressed for Model Development Efficiency

Chao Chen[1,*], Sajjad Ahmad[2], and Ajay Kalra[3]

[1,*]Department of Geosciences, Boise State University
[2]Department of Civil and Environmental Engineering and Construction, University of Nevada, Las Vegas
[3]Department of Civil and Environmental Engineering, Southern Illinois University

**Abstract.** In Coupled Groundwater and Surface-Water Flow (GSFLOW) model, the three-dimensional finite-difference groundwater model (MODFLOW) plays a critical role of groundwater flow simulation, together with which the Precipitation-Runoff Modeling System (PRMS) simulates the surface hydrologic processes. While the model development of each individual PRMS and MODFLOW model requires tremendous time and efforts, further integration development of these two models exerts addi-

tional concerns and issues due to different simulation realm, data communication, and computation algorithms. To address these concerns and issues in GSFLOW, the present paper proposes a conceptual framework from perspectives of: Model Conceptualization, Data Linkages and Transference, Model Calibration, and Sensitivity Analysis. As a demonstration, a MODFLOW groundwater flow system was developed and coupled with the PRMS model in the Lehman Creek watershed, eastern Nevada, resulting in a smooth and efficient integration as the hydrogeologic features were captured and represented. The proposed

conceptual integration framework with techniques and concerns identified substantially improves GSFLOW model development efficiency and help better model result interpretations. This may also find applications in other integrated hydrologic modelings.

## 1 Introduction

Interactions between surface water and subsurface water occur in most rivers. Depending on the hydraulic connectivity and

geologic features, the water interaction usually is complex (Scanlon et al., 2002; Winter, 2007) and affects variations in baseflow and streamflow (Ghasemizade and Schirmer, 2013; Kalra et al., 2013; Sagarika et al., 2015). This interaction may be influenced by climate, environmental factors, and human activities, resulting in spatial and temporal changes in water resources (Sophocleous, 2002; Kampf and Burges, 2007; Furman, 2008; Pathak et al., 2016; Tamaddun et al., 2016).

    Integrated hydrologic models usually are used to better understand the exchange of water between surface and subsurface

sources, interpret the water flow path, and predict water-system behavior (Kim et al., 2008; Xu et al., 2012). This type of models results from integration of a surface water system and a groundwater flow system (Prudic et al., 2015), and the coupling between surface water and subsurface flow is the core of the model (Ghasemizade and Schirmer, 2013; Carrier et al., 2016). Various algorithms and techniques are used to describe the groundwater-surface water interactions (Furman, 2008; Pathak et al.,





2016), from conceptual models (Arnold et al., 1993; Ponce et al., 1999; Osman and Bruen, 2002) to physical-based models of varying complexity (Abbott et al., 1986; Moussa et al., 2002). In recent years, more rigorous physically-based integrated models have been developed that couple one-dimensional or two-dimensional surface flow with a three-dimensional subsurface flow (Moussa et al., 2002; Kollet and Maxwell, 2006; Weill et al., 2009). For the groundwater simulation component, a three-

dimensional finite differential groundwater model developed by the U.S. Geological Survey (USGS), known as MODFLOW, has been widely used in such integrated models as SWAT-MODFLOW (Kim et al., 2008), HSPF-MODFLOW (Davis, 2001), SWAP-MODFLOW (Xu et al., 2012), TOPNET-MODFLOW (Guzha, 2008), MODHMS (Yubin Tang and Min, 2014), and GSFLOW (Markstrom et al., 2005).

The Coupled Groundwater and Surface-Water Flow (GSFLOW) model integrates the Precipitation-Runoff Modeling System

(PRMS) with MODFLOW (Harbaugh, 2005; Markstrom et al., 2005, 2015) (Fig.1), simulating both the surface hydrology and groundwater flow systems. It has been widely used in a variety of studies, such as snowmelt, surface hydrologic responses to climate change, and the effects of mining (Huntington and Niswonger, 2012; Hunt et al., 2013; Allander et al., 2014; Essaid and Hill, 2014; Hassan et al., 2014; Albano et al., 2016). Depending on the study objectives, an integrated model can operate at various temporal scales (e.g., hours, days, or months) and spatial scales (e.g., hillslope or watershed) (Goderniaux et al.,

2009; Gauthier et al., 2009; Sulis et al., 2011). This adds complexities to model development, calibration, and especially integration. Thus, it is common to simplify the model development processes by starting with decoupled surface and groundwater models. However, developing separate models without coupling concerns could result in integration challenges down the road. Extensive research efforts have focused on the coupling processes (Panday and Huyakorn, 2004), such as linking the channel flow regime with groundwater domain (Prudic, 1989; Swain and Wexler, 1996; Walton et al., 1999); linking the overland flow

with the unsaturated and saturated subsurface flow (Akan and Yen, 1981; Pinder and Sauer, 1971; Singh and Bhallamudi, 1998); and linking overland flow, channel flow, and subsurface flow to examine interactions between them (Govindaraju and Kavvas, 1991; Refsgaard and Storm, 1995). The different modeling focuses and computation algorithms in the to-be coupled models make uniqueness in the coupling procedures. However, there is very few studies are available focusing on the model development procedures of integration processes of these integrated models, such as GSFLOW model.

To fill this research gap, as the paper structure diagram shows (Fig.1), the current study proposed a conceptual model integration framework with principal concerns and issues addressed, from perspectives of: Model Conceptualization, Data Linkages and Role Transference, Model Calibration, and Sensitivity Analysis. The framework was tested and demonstrated through the development of a GSFLOW model in the Lehman Creek watershed. The subsequent modeling results were not the focus of this paper; thus no further analyses were presented beyond the model calibration and sensitivity analysis results.

While some might be interested in the performance comparisons between the PRMS model and the GSFLOW model with the MODFLOW as a component, the comparisons will be presented in authors' coming paper.

The need for this study mainly was driven by two factors. First, to develop a GSFLOW model as a coupling effort of two models, there was a need to devise techniques for a smooth and efficient transition from a stand-alone model to a component-composition model, with special attention to the interactions between two composed models, i.e., surface water and ground-

water. Second, climate change can affect local water resources; hence, an integrated understanding of the groundwater flow



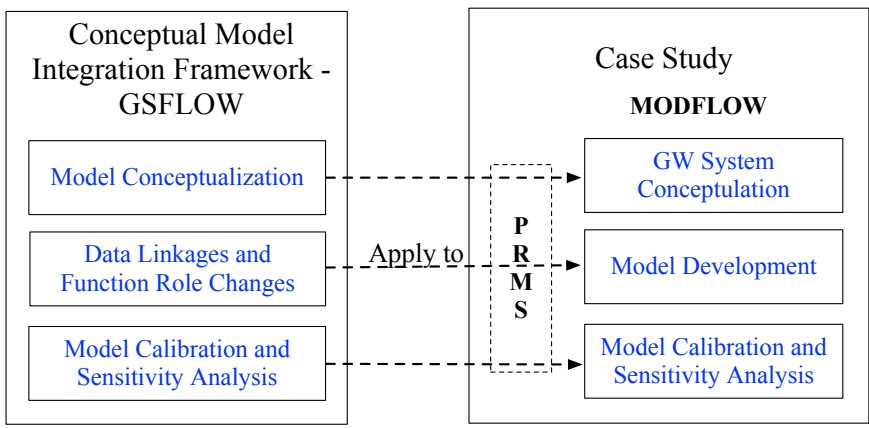

**Figure 1.** Study diagram of Conceptual Model Integration Framework with case study demonstration: couple MODFLOW with PRMS to develop GSFLOW model.

system is important in response to climate change. On the basis of available geologic conditions and hydraulic connectivity, the objective of this study was to provide techniques addressing the concerns in model integration from a perspective of MODFLOW development as a groundwater component in the GSFLOW model. The findings from this study are anticipated to provide useful information to modelers/end users regarding the integration of groundwater systems to a surface hydrologic

modeling system using GSFLOW model.

The current paper is constructed by four compartments. Firstly, the basic components and modeling scope of GSFLOW model are briefly reviewed in Section 2, and it is followed by the description of proposed conceptual framework, in Section 3, where potential concerns and issues during the development of integrated processes in GSFLOW model are identified and addressed. Then, the proposed framework is implemented and demonstrated, in Section 4, through developing a MODFLOW

model and integrating it as a groundwater model component for an fully integrated GSFLOW model. Lastly, the discussion and conclusions over the current study were made in Section5.

## 2   Overview of GSFLOW Model

GSFLOW, a Coupled Groundwater and Surface-water FLOW model, was developed by USGS (Markstrom et al., 2005), based on the integration of the Precipitation-Runoff Modeling System (PRMS) and the USGS Modular Groundwater Flow System

(MODFLOW 2005 and MODFLOW-NWT). The PRMS was developed primarily for precipitation and snowmelt runoff on





spatial-distributed physical bases, simulating processes from top of vegetative canopy to the bedrock. Based on water balance and energy balance, it particularly focuses on the surface hydrologic processes including canopy interception, snow accumulation/melt, evapotranspiration, surface runoff, and soil-water fluxes. While its groundwater flow is simplified as a stock-and-flow system, a sophisticated groundwater model would improve the modeling performance of integrated water system. MODFLOW

is a three-dimensional finite-difference groundwater flow system developed by the USGS (Markstrom et al., 2005). The finite-difference method was used to describe the spatial heterogeneity to solve groundwater flow (and contaminants) through porous mediums in three dimensions, by area (e.g., infiltration or evapotranspiration), by line (e.g., streambed infiltration and its water exchange with groundwater), or by point (e.g., water pumping and recharge). It is the most widely used simulation program for groundwater systems throughout the world (Markstrom et al., 2005). By coupling these two models, the major limitation

of each model is overcome, as the GSFLOW simulates both surface water and groundwater/subsurface-water simultaneously with dynamic water interacting through saturated and unsaturated subsurface media and through streams and lakes.

## 3 A Conceptual Framework for GSFLOW Model Integration Development

In GSFLOW, the integration script was completed by USGS, who developed both PRMS and MODFLOW model. The conceptual framework proposed herein aims to facilitate the model development of GSFLOW from a modeler perspective. Generally,

to develop a coupled GSFLOW model, the two models to be integrated are developed separately and have a pre-calibration respectively before the coupling processes (Huntington and Niswonger, 2012; Markstrom et al., 2005). Traditional model development procedures, e.g., model calibration, validation, and initialization, are applicable and required for both individual model. During these processes, different from an independent model development for non-integration purposes, there are concerns or potential issues that should be aware of or dressed, which would help modelers to better understand the GSFLOW integrated

hydrologic model, improve the efficiency of model development, and have better interpretation of simulation results. In the following sections, the main concerns or issues are addressed in the proposed framework: Model Conceptualization (section 3.1), Data Linkages and Function Role Change (section 3.2), and Model Calibration and Sensitivity Analysis (section 3.3).

### 3.1 Model Conceptualization

While two models, PRMS and MODFLOW, could have two independent approaches of model conceptualization when for

separate studies, aiming for a smooth and successful coupling development for a GSFLOW model, these two ways of model conceptualization require consistency and compatibility. It is particularly critical in terms of time and space, as they are running basics of models. During the model conceptualization, boundary definition and spatial discretization are among the most of the concerns. While the boundary of groundwater watershed is usually not as the same as surface-water watershed boundary, as the surface-water watershed is defined by topographic divides (Anderson et al., 2015) and the groundwater watershed is

not, the consistency of defined boundaries of two models should be ensured, considering the data transference between two models. During the coupling process, the groundwater simulation module in PRMS is disabled and replaced by the groundwater component MODFLOW, and the MODFLOW receives data inputs from PRMS. Any areal none-overlapped regions would be





structurally deficient with none driving inputs/forces. Especially on the level of spatially-discretized hydrologic response unit, a structure connection is required for data communication to assure vertical flows (e.g., gravity drainage) between PRMS soil zone and MODFLOW groundwater system; or else, such structure connection needs to be externally defined (Markstrom et al., 2015). Also, as temporal unit of model simulation, time step is another concern of importance. Due to different study interests,

surface water and groundwater may have different time steps in terms of hours, days, or months, depending on varied study purposes. Nevertheless, PRMS model only supports daily time step for a PRMS-IV simulation (Markstrom et al., 2015). This limits GSFLOW model simulation to a daily basis and so does the MODFLOW model component for the compatibility reason.

## 3.2 Data Linkages and Function Role Change

Leveraging the future data transferences during the model development facilitates the efficiency and effectiveness of the in-
tegrated modeling. As the groundwater component in GSFLOW model, MODFLOW interacts with the surface water system mainly through three data linkages, including:

- 1) Water percolation, resulting from the surface-water system and driving the groundwater system;

- 2) Evapotranspiration, composed by shallow ET and deep-root ET simulated by two sub-model respectively;

- 3) Streamflow, contributed by both surface runoff and dynamic water interacted with groundwater system.

**1. Water percolation - use PRMS percolation outputs as the MODFLOW driving forces for model initialization**

As driving input of groundwater model, water percolation determines groundwater system behavior and model performance. The gravity drainage, resulting from PRMS model simulation, is a portion of infiltration, after the fulfillment of shallow soil-water flow, vertically percolates into and recharges the groundwater system. The spatial distribution and value scale of magnitude of long-term percolation is determinately correlated with those of hydraulic properties in subsurface medium. As results

of PRMS surface hydrologic simulation, the value scale and spatial distribution of the gravity drainage make a well correlation between the flow rate and soil type. This well-suited correlation reflects as the driving inputs and hydraulic propertied of MOD-FLOW. Inherently, scale and use PRMS simulation gravity drainage to saves considerable efforts and time resulting a speed up for the model development in terms of initialization. Typical groundwater MODFLOW model simulation requires an initial condition set up for purposes of accurate simulation performance and a successful numerical solution (Bear, 2012; Franke

et al., 1987). Instead of initiating an independent groundwater model using numerically expensive approaches, i.e., draining test/spin-up test (Ajami et al., 2014a, b; Seck et al., 2015), directly using the PRMS model output to drive MODFLOW model initiates the data communication between models and leads a heads-up of a GSFLOW model simulation.

**2. ET – leveraging ET simulation in both PRMS and MODFLOW**

As one of the most important processes in integrated hydrologic system, ET is considered both in the soil zone of PRMS
model and the unsaturated zone of MODFLOW model. In the integrated GSFLOW model, the ET simulated in MODFLOW component represents its potential capability within the reach of deeper root depth that could not be satisfied from the soil zone simulated by the PRMS. Depending on the study purposes and hydrogeologic conditions, this could be especially important





in areas where the deep ET is active and has great influences on the seasonal variation of the water cycle. In cases while total ET were considered during the initial PRMS model development, the deep ET portion should be split out during the coupling process as to capture active variabilities. In both PRMS and MODFLOW model simulation, high variation of deep ET raises great influences in water dynamics within each sub-system, e.g., soil infiltration, soil water thresholds, soil water discharges to

or absorbed from unsaturated zone, groundwater-level, and GW storage.

### 3. Streamflow – the Role of Streamflow Routing Switches from PRMS to MODFLOW

While for both of PRMS model and MODFLOW model, the streamflow routing process is an unexclusive component regarding watershed studies. When coupling surface process with groundwater system, the function of streamflow routing process switches from by PRMS to MODFLOW, to facilitate the dynamic water interactions between streamflow and groundwater

within MODFLOW. During this switching process, concerns are from two perspectives: the routing process algorithm and the corresponding linking data. While the streamflow routing process in PRMS is simulated by Muskingum, no routing, or lake-contained algorithm, the replacing algorithms in MODFLOW require a corresponding representation of functions as was simulated in PRMS. For example, lake simulation in PRMS is part of streamflow routing process, while it is functioned by an independent module in MODFLOW. Second, stream water balance suggests data linkages between PRMS and MODFLOW:

1) the streamflow receives from /discharges to the groundwater system; 2) the overland flow that enters each stream segment. Understanding these two sources as the most critical determinant elements in the streamflow would of great help during the model calibration, which is discussed in the following section. As listed above, these three data linkages summarize the keys of simulating dynamic water interactions across the two sub-systems occurred in two critical realms: soil and stream. The smooth data communication is companied by algorithm changes with different module/packages used in both PRMS model

and MODFLOW model (Table A1,A2). Especially, the critical integration process is determined by two modules in GSFLOW (Markstrom et al., 2005; Regan et al., 2016): *gsflow_prms2mf* and *gsflow_mf2prms*. The *gsflow_prms2mf* module is used to direct PRMS outputs to MODFLOW model, which includes distributing gravity drainage and unsatisfied ET to MODFLOW and allocating surface runoff (i.e., overland flow, Dunnian runoff, and Hortonian runoff) and subsurface interflow to stream segments (Related and Tables, 2015). The *gsflow_mf2prms* module is used to distribute groundwater discharges from MOD-

FLOW cells to PRMS hydrologic response units (HRUs) when condition met. Additional parameters, which were required for these two modules, were summarized in Table A2.

### 3.3  Model Calibration and Sensitivity Analysis

Typical groundwater MODFLOW model simulation requires an initial condition set up for purposes of accurate simulation performance and a successful numerical solution (Bear, 2012; Franke et al., 1987). Instead of initiating an independent ground-

water model using numerically expensive approaches, i.e., draining test/spin-up test  (Ajami et al., 2014a, b; Seck et al., 2015), directly using the PRMS model output to drive MODFLOW model initiates the data communication between models and leads a heads-up of a GSFLOW model simulation. The gravity drainage, defined in PRMS, a portion of infiltration after the fulfillment shallow soil-water flow, vertically recharges the groundwater system. As results of PRMS model simulation, the value scale and spatial distribution of the gravity drainage make a well correlation between the flow rate and soil type. This





well-suited correlation reflects as the driving inputs and hydraulic propertied of MODFLOW and considerably saves efforts and time resulting a speed up for the model development in terms of initialization. GW model calibration is a parameterization process, which involves determining magnitude and spatial distribution of model parameters that reproduce the observed system with hydraulic heads and groundwater flows (Kim et al., 2008). Two modeling status, steady state and transient state, are both

described with MODFLOW development in the GFLOW model coupling procedure. The steady-state simulation was applied in the MODFLOW model as to set up a water balance with valid groundwater flow property, e.g., the hydraulic conductivities. The gravity drainage, deriving from the surface process model PRMS, retains valid hydraulic features of the groundwater system in terms of value scale and distribution. Such validation promotes MODFLOW simulation reaching to a steady state in the most efficient and effective way. Furthermore, by driving MODFLOW model with the gravity drainage, it initiates a

compatible communication as it will do in the eventual GSFLOW model. Any fundamental error would be easier to detect at earlier stage of a GSFLOW model development. The transient-state simulation performed in GSFLOW is with varying water flux occurring between surface-water system and groundwater system. This results parameters, engaging in the algorithms determining the rates and timing of water flux exchanges, have great influences on the integrated hydrologic system. Among all, water balance and storativities are the most concerns. While soil water changes from one-way flow simulation to two-way

flow simulation, with additional alterations in the evapotranspiration and gravity drainage computation mentioned above in Data Linkage section, the calibrated soil parameters in preliminary PRMS model requires further adjustments to reach a new soil-water balance with dynamic water flux of recharges and discharges. On the other hand, the storativity is important to the groundwater-flow simulation and may leverage the stock-featured parameters in the preliminary PRMS model to better facilitate the compatibility of two sub systems. The sensitivity analysis is an effective mean of identifying the influences of tested

parameters on the modeled system. During the transitioning from a groundwater system into an integrated hydrologic system, the influences it brings to the system changes. The highly sensitive parameters in the groundwater flow system may or may not have the similar sensitivities in the integrated system. For both stand-alone MODFLOW model and integrated GSFLOW model, the sensitivity analysis should be always performed to better understand data linkages and model behaviors, especially regarding the influences of the dynamic surface - subsurface water interactions on the integrated system.

## 4   Case Study in Lehman Creek Watershed

As in the previous studies (Chen et al., 2015, 2016), a PRMS model has been developed with detailed procedures described, aiming to evaluate the surface hydrologic responses to climate change. For a smooth and efficient transition from independent model to the integrated model, the proposed framework was applied in the following case study, with specific focuses on the development of coupling procedures in the MODFLOW.

### 4.1   Study Area

Lehman Creek watershed is located in east-central Nevada close to Nevada-Utah boundary and encompasses the Great Basin National Park (Fig.2). Defined by the surface topographical conditions, it covers an area of 23.6 km$^2$, elevating from 2040

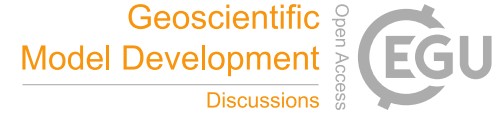



m (east) to 3980 m (west). The variable topographic reliefs and dominant coverage of evergreen forest (70.7%; (Homer et al., 2015)) make the climate dry hot at the lower plain area and the humid cool in high-elevation regions. More than 60% of the precipitation falls as snow in the mountainous areas (Volk, 2014). The Lehman Creek initiates at the glacial deposits that overlay older undifferentiated argillite, quartzite, and shale (Unp). In the cross-section shown in Fig.2, the granite and

shale intrusion separate the quartzite upstream and the limestone formation downstream, where the groundwater discharges as Cave Springs (Elliott et al., 2006). The groundwater outflows the watershed boundary, passing through the dissolute limestone formation, joining the adjacent Baker Creek (Halladay and Peacock, 1972; Elliott et al., 2006).

## 4.2   Modeling of the Groundwater Flow System

### 4.2.1   GW System Conceptualization

According to Prudic et al. (2015), in the study area where the geology is dominated by quartzite and glacial deposits (Fig.2), most of precipitation forms into surface runoff, with minor groundwater flow occurring. Groundwater flow receives a recharge from macrofractures as well as coarse sediment in the glacial deposits and alluvium with small storativities. Impervious quartzite and granite impede the groundwater flow and force it into the spring discharge (Fig.2). In the area between the intrusion and the downstream watershed boundary, the losing-stream recharges the groundwater through both glacial and al-

luvial deposits as well as the underlying karst limestone. Also, the groundwater interacts with the neighboring Baker Creek watershed at southeastern boundary (Prudic et al., 2015). To couple the MODFLOW with surface hydrologic PRMS model in a simple and straightforward approach, the identical modeling area and grid mash as used in the PRMS model were applied in the MODFLOW model to ensure the data communication between two sub-systems on both region level and grid level. Yet, it resulted in adjustments in boundary conditions to compensate the imbalanced water cut-off due to the different

"watershed" definitions in surface water and groundwater system. Herein, the spring discharges and the groundwater outflows were considered on the basis of water balance estimation, as the boundary conditions. As the Table 1 shows, the water balance estimation includes vertical infiltration as inflow (1010 m$^3$/d), derived from the PRMS model; the system outflows of baseflow (450 m$^3$/d; (Prudic et al., 2011, 2015)), spring discharge (245 m$^3$/d; (Halladay and Peacock, 1972; Prudic and Glancy, 2009)), and groundwater outflows at an estimation of 315 m$^3$/d. Fig. 3 shows the position where the boundary flux occurred. A two-

layer groundwater flow system was defined, based on hydrogeologic features (Maxey, 1964; Seaber, 1988). Layer 1 consisted of glacial and alluvial deposits and Layer 2 consisted of fractured quartzite at the upstream, limestone at the downstream, split by granite and shale intrusions (Fig.3). The granite and shale that underlie the fractured quartzite only was represented at the intrusion as model bottom was considered as no-flow boundaries in this model (Fig.3).

### 4.2.2   Model Development

Apart from the fundamental MODFLOW model development, e.g., model package setup and parameterization (Chen et al., 2017), concerns from Data Linkages and Function Role Change were addressed specifically. While in the previous Lehman Creek PRMS model development (Chen et al., 2015), the evapotranspiration is over-estimated as to represent the groundwater





**Figure 2.** (a) The surface geology map in the Lehman Creek watershed, Great Basin National Park, Nevada and (b) an interpretive geologic cross-section with location indications for Cave Springs and Lehman Caves (adapted from (Prudic et al., 2015)).



**Figure 3.** (a) The surface geology map in the Lehman Creek watershed, Great Basin National Park, Nevada and (b) an interpretive geologic cross-section with location indications for Cave Springs and Lehman Caves (adapted from (Prudic et al., 2015)).





**Table 1.** Water Budget Estimations of the Conceptualized Groundwater Flow System in the Lehman Creek Watershed under Steady-State Simulation.

| | Water Budget Component | Flow Rate (m³/d) | Estimation Source |
|---|---|---|---|
| Inflow | Vertical infiltration | 1010 | Water balance estimation |
| Outflow | Streamflow baseflow | 450 | Measurements and Prudic et al. (2015) |
| | Spring discharge | 245 | Prudic and Glancy (2009) |
| | Groundwater flow | 315 | Estimation from Prudic et al. (2015) |

**Table 2.** Hydraulic conductivity of each hydrostrategraphic unit in the MODFLOW model (unit: m/d).

| Hydrostrategraphic Unit | Horizontal | Vertical | Value Ranges of selected rocks (Heath 1983) | |
|---|---|---|---|---|
| Glacial deposits | 5.00E-02 | 2.20E-04 | Glacial Till | 1E-7 to 3E-1 |
| Alluvial deposits | 5.00E-02 | 1 | Silty, Loess, Silty Sand, Clean Sand, Gravel | 1E-3 to 5E3 |
| Quartzite | 5.00E-07 | 5.00E-07 | Igneous and Metamorphic Rock | 1E-8 to 5 |
| Limestone | 1.00E-04 | 2.50E-01 | Carbonate Rocks | 1E-4 to 5E3 |
| Granite and Pioche shale | 1.00E-07 | 1.00E-07 | Shale | 1E-8 to 1E-4 |

loss in the water balance, which includes the Cave Spring and groundwater outflows to the adjacent Baker Creek drainage (Volk, 2014). The parameter (*jh_coef*) determining the potential evapotranspiration in the soil was adjusted with a reduction, and the compensation was made by deep-root evapotranspiration simulated by the MODFLOW. Secondly, the gravity drainage from PRMS was the MODFLOW model-driving inflow and was in balance with groundwater outflows that were not considered in

the PRMS model, including spring discharges and boundary outflows, by adjusting the parameter (*ssr2gw_rate*, *ssr2gw_exp*) to modify the exponential curve that determining the gravity drainage rate. In terms of role exchange, all the routing processes and related parameters in the previous PRMS model were forfeited, and a new module, as a role replacement in the integrated GS-FLOW model, StreamFlow Routing packages (SFR) was used to present the streamflow routing process from stream originate to the outlet of the watershed and to account for the streamflow-groundwater interactions. Streambed thickness and hydraulic

conductivity were estimated for each specified featured hydrogeologic formation, according to piezometer measurements and literature studies (Prudic and Glancy, 2009; Allander and Berger, 2009) Detailed changes in modules and related parameters can be found in appendix.

### 4.2.3   Model Calibration and Sensitivity Analysis

The calibration procedure for MODFLOW, as a component of an integrated hydrologic model GSFLOW, includes steady-state

and transient-state model calibration, which were performed for both MODFLOW_only simulation and integrated GSFLOW simulation separately. In this study, the model was calibrated using a trial-and-error technique for both simulations, to esti-

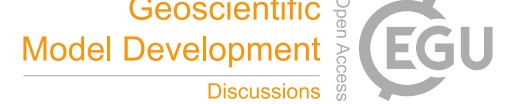

mate the hydraulic conductivity and the storativity of each hydrostratigraphic unit by fitting the water budget results with the estimations within a ±10% range (Table 1). Due to the observation shortage as there were no drill holes or well observations within the region of the study area, hydrogeologic features and components in water balance were used for the steady-state model calibration; the streamflow observations were used for the transient-state model calibration. The sensitivity analysis was

carried out to estimate hydraulic conductivity of each hydrostrategraphic unit to assess the influences on the water-balance estimation. The model simulations were conducted for 14 different values of hydraulic conductivity, ranging from 0.2 to 10 times the estimated value. For each run, the Root Mean Square Error (RMSE) was calculated, which measures the error of fitness of the estimation to the data (Kenney John, 1939). The lower value of RMSE meant a better model simulation with fewer errors. Accordingly, the sensitivity of the hydraulic conductivity values could be estimated by corresponding changes in

the model errors.

### 4.3 Modeling Results

By applying the conceptual framework and having potential concerns addressed, the MODFLOW development playing a compotential role in the integrated system turns out successful, as the transition from an independent model to a system component was smooth, efficiently, and effectively. Apart from continuous converging computation in MODFLOW and GSFLOW, the re-

sulting hydraulic property estimations and their influences on the integrated system well represent the geophysical conditions and hydrogeophysical features in the study area. The coarse glacial deposits sitting in the central valley and along the streams has a relative high hydraulic conductivity (5E-2 m/d horizontally and 3E-2 m/d vertically); they had the most influences on the modeling performance, as the greatest increase was found in the water balance RMSE results when they varied. Around the downstream side, the alluvial deposits had a higher vertical hydraulic conductivity (1 m/d) than horizontal (5E-2 m/d), which

fits with the losing-stream feature observed. Underneath them, the fractured Prospect mountain quartzite and the granite (and Pioche shale) intrusion had a low hydraulic conductivity of 5E-7 m/d and 1E-7 m/d, respectively, with the least effects on the integrated system as the RMSEs did not response much when the parameters changed; the limestone unit where forms the Lehman Cave had high as hydraulic conductivities of 4E-4 m/d vertically and 1E-2 m/d horizontally. The model simulation showed groundwater discharged where the glacial deposits meet the Granite and Pioche shale as springs. Overall, as RMSE is

an absolute measure of fitness, the current model error was 31 $m^3$/d with calibrated parameters (Table 2).

### 5 Discussion and Conclusion

The primary objective of the current study is to propose a conceptual integration framework with techniques and concerns identified and addressed that can improve GSFLOW model development efficiency and help better simulation interpretations. Focusing on the main elements in modeling procedures, Model Conceptualization, Data Linkage and Function Role

Change, and Calibration and Sensitivity Analysis, the proposed conceptual framework identified the keys for a successful model communication between two sub-models, i.e., PRMS and MODFLOW, within GSFLOW model. The tackling strategies and techniques were proposed correspondingly. As a demonstration, the proposed framework was applied to a study in the





Lehman Creek watershed. As the example demonstrates how to implement the conceptual framework from the perspective of the MODFLOW model development, it also showed how to initialize and modify the model, together with the PRMS model, using the proposed techniques for being a model component in GSFLOW model. After model calibration, the modeling results well estimated the hydraulic conductivities and storativities of the defined stratigraphic units, which kept the water balance

estimation and captured the hydrogeologic features with spring discharges and groundwater outflows. In this study, the main conclusions drawn from this study are:

– Keeping a consistency in spatial and temporal discretization of two sub systems is important to the GSFLOW model development, while such consistency restrains the implementation of GSFLOW model due to temporal scale and raises extra requirements for boundary conditions due to spatial definition differences;

– Leveraging three active data linkages, vertical percolation, deep-root ET uptake, and streamflow-aquifer interactions, in the integrated model development is critical for successful data communication and subsequent dynamics within two sub system and inherently the integrated system;

– Using the Gravity Drainage result of PRMS model to drive MODFLOW model is an efficient technique to: 1) fast converge the groundwater modeling as it keeps the soil texture in surface hydrologic simulation align with hydraulic

properties in groundwater system simulation; 2) debug the initialized GSFLOW at its early-stage;

– Applying the proposed conceptual framework is practically useful for an effective and efficient GSFLOW model development.

This research sets up the fundamentals of GSFLOW model development, focusing on the transitioning processes developing a stand-along model into a model component. Based on current documented literatures and authors' knowledge, this

is the first study providing modeling efficiency strategies of GSFLOW model development with better understanding of the model integration structure and algorithms, as it identified the key concerns and issues regarding modeling scopes in and data communication between the two sub-models. Although authors just present the basic model development outcomes, in terms of model calibration and sensitivity analysis, there is already a substantial body of studies with detailed results and research findings while missing illustrations of procedures and techniques in the model development. To know such missing procedures

is appealing: it offers a view of data communication between two sub-models from a perspective other than a vague concept of integrated system, and thus, when reconciled with model development perspective, can help obtain a more coherent image. The proposed framework inherently provides additional hands-on guidance of a GSFLOW model development apart from its manual. Additionally, to other integrated hydrologic modelers, this study also provides valuable experiences where common concepts are shared.

*Code and data availability.*  The GSFLOW (v1.2.2) is open sourced program and its code and related documentation are available on USGS web page: https://water.usgs.gov/ogw/gsflow/. While being compositions of GSFLOW program, both PRMS and MODFLOW are indepen-




dent open-sourced program. The code and related documentation can be found through: https://wwwbrr.cr.usgs.gov/projects/SW_MoWS/
PRMS.html and https://water.usgs.gov/ogw/modflow-nwt/

*Author contributions.* SA initiated this research and conceived the study. CC carried it out with model development and simulations. CC and
AK discussed and designed the workflow. CC prepared the manuscript with contributions from all co-authors.

*Acknowledgements.* We would like to acknowledge Karl Pohlman, Tracie Jackson, Lynn Fenstermaker, and Chang Liao for providing suggestions, advice, and help for the model development. USGS report by Prudic et al. (2015) was very helpful in understanding the geological
formation of the study area and estimating the model parameters. We also want to thank Miguel Aguayo and Lejo Flores for all the support
and help for the paper submission. This work was supported by NSF under Grant IIA-1329469.

**Abbreviation** The following abbreviations are used for hydrologic models mentioned in this manuscript.

– GSFLOW Coupled groundwater and surface-water flow model

  – PRMS Precipitation-Runoff Modeling System

  – MODFLOW Modular three-dimensional (3D) finite-difference groundwater model

  – MODFLOW-NWT A model that uses the Newton-Rapshon formulation for MODFLOW-2005

  – HSPF Hydrological simulation program - Fortran

– MODHMS A comprehensive MODFLOW-based hydrologic modeling system

  – SWAP Soil Water Atmosphere Plant

  – SWAT Soil and Water Assessment Tool

  – TOPNET Networked version of a topographic model

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
