# Peer review of "A Conceptual Framework for Integration Development of GSFLOW Model: Concerns and Issues Identified and Addressed for Model Development Efficiency"

_Geoscientific Model Development, 2018_

## Short Comment (SC1) · 18 Dec 2018

Dear authors,

In my role as Executive editor of GMD, I would like to bring to your attention our Editorial version 1.1:

http://www.geosci-model-dev.net/8/3487/2015/gmd-8-3487-2015.html

This highlights some requirements of papers published in GMD, which is also available

on the GMD website in the 'Manuscript Types' section:

http://www.geoscientific-model-development.net/submission/manuscript_types.html

In particular, please note that for your paper, the following requirements have not been met in the Discussions paper:

- "The main paper must give the model name and version number (or other unique identifier) in the title."

- "All papers must include a section, at the end of the paper, entitled 'Code availability'. Here, either instructions for obtaining the code, or the reasons why the code is not available should be clearly stated. It is preferred for the code to be uploaded as a supplement or to be made available at a data repository with an associated DOI (digital object identifier) for the exact model version described in the paper. Alternatively, for established models, there may be an existing means of accessing the code through a particular system. In this case, there must exist a means of permanently accessing the precise model version described in the paper. In some cases, authors may prefer to put models on their own website, or to act as a point of contact for obtaining the code. Given the impermanence of websites and email addresses, this is not encouraged, and authors should consider improving the availability with a more permanent arrangement. After the paper is accepted the model archive should be updated to include a link to the GMD paper."

Thus add the version number of GSFLOW to the title of your manuscript.

GMD is encouraging authors to provide a persistent access to the exact version of the source code used for the model version presented in the paper. As explained in https://www.geoscientific-model-development.net/about/manuscript_types.html the preferred reference to this release is through the use of a DOI which then can be cited

in the paper. In case your institution does not provide the possibility to make electronic data accessible through a DOI you may consider other providers (eg. zenodo.org of CERN) to create a DOI. Please note that in the code accessibility section you can still point the reader to the repository for the newest version even if you use a DOI for the relevant releases.

Yours,

Astrid Kerkweg

---

## Referee Comment (RC1) · Anonymous Referee #1 · 22 Dec 2018

The presented study promises a "conceptual framework from perspectives of: Model Conceptualization, Data Linkages and Transference, Model Calibration, and Sensitivity Analysis" for an existing model GSFLOW. GSFLOW is a model developed by the USGS, which couples two existing models: PRMS and MODFLOW. The framework is intended to improve GSFLOW model development efficiency and help the interpretation of simulation results.

It is not clear how the presented study adds information to the community not already available in the USGS documentation of the used model frameworks. The authors

claim that they present a conceptual integration framework which is only outlined by one figure. It is unclear what a conceptual integration framework constitutes and why it entails benefits for the interpretation of simulation results yet alone model development efficiency.

The presented study builds on already available code and does not present any new information about the GSFLOW model. A structured guideline for modelers would help the community to implement these kinds of models and is promised by the authors but not presented in any way in the paper. The presented study presents a tutorial that could be uploaded to a publication platform like github to provide an example study for the community but does not fit the requirements for a GMD development and technical paper. No insides to technical aspects to running models or reproducibility of results are presented. The GMD guidelines state that: "Development and technical papers usually include a significant amount of evaluation against standard benchmarks, observations, and/or other model output as appropriate." The presented model of the Lehman Creek watershed was already described in a different publication and does not add any of the evaluation mentioned above. It would be appropriate if the authors used the model to show how the transition from an independent model to the integrated model is supported by their "framework". At this point I cannot see any evidence of a detailed evaluation or discussion on this topic. The shown figures are adapted from a previous publication and do not add any valuable information in evaluating the proposed framework. Additionally, these figures do not meet the standards for a GMD publication.

Additional remarks: - Unclear use of language, missing articles or wrong use of plural - Units do not follow journal guidelines - Figures text is not aligned and partially unreadable (e.g. Fig 2b))

---

## Referee Comment (RC2) · Anonymous Referee #2 · 25 Dec 2018

The authors purport to provide insights that guide the use of the USGS's integrated hydrologic model GSFLOW, and they then present an example implementation. Unfortunately, I cannot recommend this manuscript for publication in GMD. First, the article needs substantial editing for English grammar, which makes it difficult to read. Second, I have difficulty seeing what is the new contribution in this paper. The abstract says: "the present paper proposes a conceptual framework from perspectives of: Model Conceptualization, Data Linkages and Transference, Model Calibration, and Sensitivity Analysis." This is extremely vague. The capitalization of these generic concepts gives the reader the impression that they authors will provide a unique, new idea for each of these. However, when these are explained in detail in Sections 3.1-3.3, I do not see more than a summary of how GSFLOW works. In particular, Section 3.1 Model Conceptualization simply describes the different domains covered by the model. Section 3.2 Data Linkages and Function Role Change reads like a brief summary of the model manual on how the different domains are linked. I thought that Section 3.3 Model Calibration and Sensitivity Analysis could be the place where there is something new – maybe the authors had developed a model calibration or sensitivity analysis method to be added to GSFLOW, but instead, I have a hard time understanding what the authors are discussing here (just pointing out what parameters should be adjusted?), and they themselves just use "trial-and-error" calibration in their example.

This manuscript seems to read a bit like a report of how the authors figured out and applied GSFLOW, with no new additions to the model or its implementation process. Given that GSFLOW is an already developed and published model, with numerous implementations in the literature, just the authors' ability to run it does not seem to merit a new publication. They possibly could have made a contribution through a new interpretation of their simulation findings, but they explicitly state that they will not explain their results, because they want to save this for a later paper. Their Summary and Conclusion provides 4 bullet points that are their main takeaways. The first is a finding that consistent discretizations between the 2 sub-domains is important – this is the only research-like finding that I saw in the manuscript, but I don't actually see this claim demonstrated in the manuscript. The second two points seem to only summarize how the sub-domains of the GSFLOW are linked – which is just about GSFLOW and not anything new done by the authors. Lastly, they state that their proposed conceptual framework is effective, but I'm afraid I do not see what new is proposed. I might also point the authors to 2 recent papers on software packages that aid users in the implementation of GSFLOW:

Gardner, M. A., Morton, C. G., Huntington, J. L., Niswonger, R. G., & Henson, W.

R. (2018). Input data processing tools for the integrated hydrologic model GSFLOW. Environmental Modelling & Software, 109, 41-53.

Ng, G. H., Wickert, A. D., Somers, L. D., Saberi, L., Cronkite-Ratcliff, C., Niswonger, R. G., & McKenzie, J. M. (2018). GSFLOW–GRASS v1. 0.0: GIS-enabled hydrologic modeling of coupled groundwater–surface-water systems. Geoscientific Model Development, 11(12), 4755-4777.

---

## Referee Comment (RC3) · Anonymous Referee #3 · 12 Jan 2019

After reading the entire paper, I found there are serious fundamental flaws in the organization and perhaps the key concepts of this paper, as I will detail below. Therefore, I cannot recommend the publication of this paper.

Regarding the structure of this paper, the following serious shortcoming can be clearly seen -

1. The "Modeling Results" of this study, which is supposed to be the most important, is only one paragraph in Section 4.3.

2. In contrast, too many general statements written in a very lengthy way. For example, in Section 3.2 the three key data linkages are all common knowledge which i am sure most researchers are well aware of these basic processes. There is really not worthwhile to spend so many pages in explaining them while the important Result section is kept minimal.

(Note that A paragraph in Section 3.3 is 31 lines long. It is very tough for any readers to follow.)

3. There are only three Figures - two of them were borrowed from another paper. The remaining one figure is also not related to the results of this paper. There are two Tables, but I cannot see why they are necessary to be presented. Why and how can there are no figures/tables at all on the modeling results?

4. Too many long and repeated sentences which make super long paragraphs over and over. For example, a paragraph in Section 3.3 is 31 lines long. It is very tough for any readers.

As one key achievement and conclusion of this study, the authors stated that (P12, L12-14) in Section 4.3 "Modeling Results" (which is way too short !! only one paragraph, 14 lines for the results!)

"the MODFLOW development playing a componential role in the integrated system turns out successful, as the transition from an independent model to a system component was smooth, efficiently, and effectively.

But, I do not think the authors have demonstrated what they said in this sentence at all. What does those adjectives "smooth, efficiently, and effectively" really mean? It is very vague, and, how to measure them? also they are "smooth, efficient, and effectively" relative to which previous modeling systems?

Regarding the key concepts, I cannot agree that all the contents in Section 3.1 are true. All these statements are too general rather than being specific for the certain situation. I

think the authors were promoting the dynamic coupling between surface water systems and groundwater system, for example, they stated "... such structure connection needs to be externally defined....". However, I cannot fully agree. I believe in some situations, a data transfer (e.g. groundwater recharge) simulated from one model passing to the other can well suit the situation without interaction. It really depends on the hydrologic settings, but it was pity that the authors did not spell out the entire picture, i..e, when the fully interactive coupling is necessary, and if the coupling is necessary, how to take care of water balance issues in two coupled models to conserve water? All of these key issues seem like not being focused in this paper.

Finally, there are also many editorial revisions required to be made throughout the entire manuscript. I summarize some obvious ones below for the authors' reference.

Editorial Comments -

P2 L22: Revise –

"in the to-be coupled models make uniqueness in the coupling procedures..."

Into -

"in the models to be coupled make uniqueness in the coupling procedures..."

P2 L23, L24: Revise –

"However, there is very few studies are available focusing on the model development procedures of integration processes of these integrated models,..."

Into -

"However, there is very few studies available focusing on the model development procedures of these integrated models,..."

P2 L26: Revise –

"with principal concerns and issues addressed, from perspectives of:..."

into -

"with the principal concerns and issues addressed, from the perspectives of..."

P4 L24-25: Revise "when for separate studies" P5 L22: Revise the entire sentences of "scale and use PRMS simulation gravity drainage to saves considerable efforts and time resulting a speed...". Some grammar errors here, and not easy to understand.

P6 L2: Revise the wording "deep ET portion" by referring to the process instead of shallow or deep. P6 L7: Revise the entire sentence here. It does not read like a sentence to start with "While".

P6 L9: Revise "switches from by PRMS to MODFLOW" P6 L28: Revise "Typical groundwater MODFLOW model simulation requires an initial..."

P6 L33-34 Revise "after the fulfillment shallow soil-water flow....". What is "value scale"? You mean the "magnitude"?

P12 L17: Revise "5E-2 m/d" since it was seldom written in this way for any papers.